# Convergence Gain in Compressive Deconvolution: Application to Medical Ultrasound Imaging

**Bin Gao [1,2], Shaozhang Xiao [1], Li Zhao [2], Xian Liu [3,4,*] and Kegang Pan [3,4]**

[1] Faculty of Computer and Software Engineering, Huaiyin Institute of Technology, Huaiyin 223003, China; feimaxiao123@gmail.com (B.G.); 11040034@hyit.edu.cn (S.X.)

[2] Key Laboratory of Underwater Acoustic Signal Processing of Ministry of Education, Southeast University, Nanjing 210096, China; zhaoli@seu.edu.cn

[3] College of Communications Engineering, Army Engineering University of PLA, Nanjing 210007, China; 13701581949@163.com

[4] Key Laboratory of PLA Military Satellite Communication, Nanjing 210007, China

* Correspondence: 13584013508@163.com

**Abstract:** The compressive deconvolution (CD) problem represents a class of efficient models that is appealing in high-resolution ultrasound image reconstruction. In this paper, we focus on designing an improved CD method based on the framework of a strictly contractive Peaceman–Rechford splitting method (sc-PRSM). By fully excavating the special structure of ultrasound image reconstruction, the improved CD method is easier to implement by partially linearizing the quadratic term of subproblems in the CD problem. The resulting subproblems can obtain closed-form solutions. The convergence of the improved CD method with partial linearization is guaranteed by employing a customized relaxation factor. We establish the global convergence for the new method. The performance of the method is verified via several experiments implemented in realistic synthetic data and in vivo ultrasound images.

**Keywords:** linearized Peaceman–Rechford splitting method; compressive deconvolution; convex minimization; compressive sensing

---

## 1. Introduction

Ultrasound imaging has become a very important medical imaging scheme, as it is noninvasive, harmless, and cost-effective compared with computed tomography (CT) and nuclear magnetic resonance (MRI).

Moreover, ultrasound requires little energy, which makes ultrasound imaging a good candidate for hand-held applications. An extremely promising application scenario for ultrasound imaging is breast cancer localization, where cancer regions have statistically different acoustic properties compared with benign areas [1].

Compressive sensing (CS) can accelerate the acquisition rate without decreasing the reconstructed signal quality and maintain the image quality with fewer data. Based on CS theory, one can effectively implement the reconstruction under the condition of the restricted isometry property (RIP) [2,3]. The applicability of CS, such as 2D and 3D ultrasound imaging [4] or duplex Doppler [5], has attracted an increasing number of researchers to propose new theories and methods. Among them, several ultrasound imaging devices [6–10] have been proposed in which acquisition and compression are processed simultaneously. The main idea is to combine CS and deconvolution into ultrasound imaging and form the compressive deconvolution problem as follows:

$$y = \Phi H x + \nu, \tag{1}$$

where $y \in \mathbb{R}^M$ involves $M$ linear measurements acquired by projecting one RF image $Hx \in \mathbb{R}^N$ onto the CS acquisition matrix $\Phi \in \mathbb{R}^{M \times N}$, with $M \ll N$. $H \in \mathbb{R}^{N \times N}$ represents a block circulant with a circulant block (BCCB) matrix modeling the 2D convolution between the 2D point spread function (PSF) of the ultrasound system, and the tissue reflectivity function (TRF), $\nu \in \mathbb{R}^M$, represents a zero-mean additive white Gaussian noise.

Some sequential approaches such as YALL1 [11] are prone to first reformulate (1) into an unconstrained optimization model:

$$a = \arg\min_a \{\|y - \Phi\Psi a\|_2^2 + 2\mu\|a\|_1\}. \tag{2}$$

When the blurred RF image $\tilde{r} = \Psi a$ is restored, the TRF $x$ will be inferred by minimizing:

$$\min_{x \in \mathbb{R}^N} \alpha\|x\|_1 + \|\tilde{r} - Hx\|_2^2. \tag{3}$$

The above two-step manipulation is the most intuitive way but inevitably yields larger estimation errors as illustrated in [12]. As such, Chen et al. [13] investigate the following model by solving CS and deconvolution problems simultaneously:

$$\min_{x,a \in \mathbb{R}^N} \|w\|_1 + \alpha\|x\|_q^q + \frac{1}{2\mu}\|y - Aa\|_2^2 \tag{4}$$

$$\text{s.b.} \quad Hx = \Psi a, w = a, A = \Phi\Psi$$

where $q$ denotes the shape parameter of the generalized Gaussian distribution.

Define $A_1 = \begin{pmatrix} I_N \\ \Psi \end{pmatrix}$, $B_1 = \begin{pmatrix} -I_N & 0 \\ 0 & -H \end{pmatrix}$, $\lambda = \begin{pmatrix} \lambda_1 \\ \lambda_2 \end{pmatrix}$ and $p = \begin{pmatrix} w \\ x \end{pmatrix}$, Equation (4) can be solved by alternatively solving three easier sub-problems based on alternating direction method of multipliers (ADMM):

$$w^{k+1} = prox_{\|\cdot\|_1/\beta}\left(a^k - \frac{\lambda^k}{\beta}\right), \tag{5a}$$

$$x^{k+1} \approx prox_{\alpha\gamma\|\cdot\|_p^p/\beta}\left(x^k - \gamma H^T(Hx^k + \frac{\lambda_2^k}{\beta} - \Psi a^k)\right), \tag{5b}$$

$$a^{k+1} = (\frac{1}{\mu}A^T A + \beta I_N + \beta\Psi^T\Psi)^{-1}(\frac{1}{\mu}A^T y + \lambda_1^k + \Psi^T\lambda_2^k + \beta w^{k+1} + \beta\Psi^T Hx^{k+1}), \tag{5c}$$

$$\lambda^{k+1} = \lambda^k - \beta(A_1 a^{k+1} - B_2 p^{k+1}) \tag{5d}$$

where $h(x) = \frac{1}{2}\|\Psi a^k - Hx - \lambda_2^k\|_2^2$, $\gamma$ is a Lipschitz parameter, and *prox* represents the proximal operator [14]. For the above three sub-problems, $w^{k+1}$ has a closed solution and $x^{k+1}$ has an approximate closed solution, but $a^{k+1}$ is not easy to compute, even though the effective Sherman–Morrison–Woolbury inversion manipulation can be introduced. For non-orthogonal sparse basis $\Psi$, approximation algorithms such as Newton's method are quite time-consuming [15]. In terms of saving such computing time, some accelerating strategies and new iterative schemes need to be investigated and explored. Meanwhile, since the compressive deconvolution (CD) method is an inexact ADMM by solving $x^{k+1}$ approximately, it is extremely important to present a strict and complete convergence analysis.

Motivated by these observations and based on the semi-proximal Peaceman–Rechard splitting method in our previous work [16], in this paper, we propose an improved CD method, referred to as semi-proximal symmetric ADMM. Define $v = \Psi a$, and the improved compressive deconvolution (ICD) method has the following iterative scheme:

$$w^{k+1} = prox_{\tau\|\cdot\|_1/\beta}\left(v^k - \tau\Psi^{\mathrm{T}}(\Psi w + \tfrac{\lambda_1^k}{\beta} - v^k)\right), \tag{6a}$$

$$x^{k+1} = prox_{\alpha\gamma\|\cdot\|_p^p/\beta}\left(x^k - \gamma H^{\mathrm{T}}(Hx^k + \tfrac{\lambda_2^k}{\beta} - v^k)\right), \tag{6b}$$

$$\lambda^{k+\frac{1}{2}} = \lambda^k - \rho\beta(A_2 v^k - B_2 p^{k+1}), \tag{6c}$$

$$v^{k+1} = (\tfrac{1}{\mu}\Phi^{\mathrm{T}}\Phi + 2\beta I_N)^{-1}(\tfrac{1}{\mu}\Phi^{\mathrm{T}}y + \lambda_1^k + \lambda_2^k + \beta\Psi w^{k+1} + \beta Hx^{k+1}), \tag{6d}$$

$$\lambda^{k+1} = \lambda^{k+\frac{1}{2}} - \varrho\beta(Av^{k+1} - Bp^{k+1}), \tag{6e}$$

where $A = \begin{pmatrix} I_N \\ I_N \end{pmatrix}$, $B = \begin{pmatrix} -\Psi & 0 \\ 0 & -H \end{pmatrix}$, $\rho \in (0,1)$ and $\varrho \in (0,1]$.

Compared with the CD method presented in [13], the contributions of this article can be summarized as follows:

1.  Based on the iterative scheme of the strictly semi-proximal Peaceman–Rechard splitting method, we present an ICD method that will reduce the number of iterations while only involving additional dual update (i.e., $\lambda^{1+\frac{1}{2}}$) and requiring almost the same computational effort for each iteration.
2.  We prove that the ICD method will converge under mild conditions, while the convergence analysis is not given in the previous CD method.
3.  We introduce some elaborate manipulations that can directly generalize the CD method to more general scenarios with a non-orthogonal sparse basis $\Psi$.

The rest of this paper is organized as follows. In Section 2, we first present some preliminaries that are useful for subsequent analysis. Then, we illustrate the ICD method to rebuild the sparse coefficients from the measurements of ultrasound imaging, and the convergence analysis is given. In Section 3, we give extensive ultrasound experiments that can be used to evaluate the performance of the proposed reconstruction algorithm in comparison with CD algorithm. Finally, we make some concluding remarks in Section 4.

## 2. Method

### 2.1. Preliminaries

#### 2.1.1. Variational Reformulation of Equation (4)

In this section, inspired by He and Yuan's approach [17], we equivalently convert the convex minimization model expressed by Equation (4) to a variational form. It makes sense to perform such reformulation, because convergence analysis becomes more concise under the variational model.

For succedent analysis of the proposed algorithm, let us denote $z_1 = v = \Psi a$ and $z_2 = p = \begin{pmatrix} w \\ x \end{pmatrix}$. Then the ultrasound imaging model can be reformulated as

$$\min_{z_1, z_2 \in \mathbb{R}^N} \theta_1(z_1) + \theta_2(z_2) \tag{7}$$

$$s.b. \quad Az_1 + Bz_2 = 0,$$

where $\theta_1(z_1) = \frac{1}{2\mu}\|y - \Phi z_1\|_2^2$, $\theta_2(z_2) = \|w\|_1 + \alpha\|x\|_p^p$. The Lagrangian function and augmented Lagrangian function of Equation (7) can be, respectively, expressed as

$$\mathcal{L}(z_1, z_2, \lambda) = \theta_1(z_1) + \theta_2(z_2) - \lambda^{\mathrm{T}}(Az_1 + Bz_2) \tag{8}$$

and

$$\mathcal{L}_\beta(z_1, z_2, \lambda) = \theta_1(z_1) + \theta_2(z_2) - \lambda^\mathsf{T}(Az_1 + Bz_2) + \frac{\beta}{2}\|Az_1 + Bz_2\|^2, \tag{9}$$

where $\lambda \in \mathbb{R}^N$ represents the Lagrangian multiplier. Then hunting for a saddle point of $L(z_1, z_2, \lambda)$ is to seek $(z_1^*, z_2^*, \lambda^*)$ such that

$$\mathcal{L}_{\lambda \in \mathbb{R}^N}(z_1^*, z_2^*, \lambda) \leq \mathcal{L}(z_1^*, z_2^*, \lambda^*) \leq \mathcal{L}_{z_1 \in \mathbb{R}^N, z_2 \in \mathbb{R}^N}(z_1, z_2, \lambda^*). \tag{10}$$

That is, for any $(z_1, z_2, \lambda)$, we have

$$\theta_1(z_1) + \theta_2(z_2) - (\theta_1(z_1^*) + \theta_2(z_2^*)) - (z_1 - z_1^*)^\mathsf{T} A^\mathsf{T} \lambda^* - (z_2 - z_2^*)^\mathsf{T} B^\mathsf{T} \lambda^* \geq 0, \tag{11a}$$

$$(\lambda - \lambda^*)^\mathsf{T}(Az_1 + Bz_2) \geq 0. \tag{11b}$$

Then, resolving Equation (4) is equivalent to seeking $w = (z_1^*, z_2^*, \lambda^*)$ such that

$$\text{VI}(\Omega, F, \theta) : \theta(u) - \theta(u^*) + (w - w^*)^\mathsf{T} F(w^*) \geq 0, \forall w \in \Omega \tag{12}$$

where

$$u = \begin{pmatrix} z_1 \\ z_2 \end{pmatrix}, \quad w = \begin{pmatrix} z_1 \\ z_2 \\ \lambda \end{pmatrix}, \quad \theta(u) = \theta_1(z_1) + \theta_2(z_2) \quad \text{and} \quad F(w) = \begin{pmatrix} -A^\mathsf{T}\lambda \\ -B^\mathsf{T}\lambda \\ Az_1 + Bz_2 \end{pmatrix}. \tag{13}$$

Especially, the mapping $F(w)$ defined in Equation (13) is affine with a skew-symmetric matrix, it is monotone. We express by $\Omega^*$ the solution set of $\text{VI}(\Omega, F, \theta)$.

### 2.1.2. Notations

We denote the 2-norm of a vector by $\|\cdot\|$ and let $\|z\|_G^2 = z^\mathsf{T} G z$ for $z \in \mathbb{R}^N$ and $G \in \mathbb{R}^{N \times N}$. For a real symmetric matrix $S$, $S \succeq 0$ ($S \succ 0$) represents $S$, which is positive semidefinite (positive definite). For ease of the analysis, we define the following matrices as

$$H = \begin{pmatrix} R & 0 & 0 \\ 0 & \frac{\rho + \varrho - \rho\varrho}{\rho + \varrho}\beta B^\mathsf{T} B & -\frac{\rho}{\rho + \varrho}B^\mathsf{T} \\ 0 & -\frac{\rho}{\rho + \varrho}B & \frac{1}{(\rho + \varrho)\beta}I_N \end{pmatrix} \tag{14}$$

$$M = \begin{pmatrix} I_n & 0 & 0 \\ 0 & I_N & 0 \\ 0 & -\varrho\beta B & (\rho + \varrho)I_N \end{pmatrix} \tag{15}$$

and

$$Q = \begin{pmatrix} R & 0 & 0 \\ 0 & \beta I_N & -\rho B^\mathsf{T} \\ 0 & -B & \frac{1}{\beta}I_N \end{pmatrix}. \tag{16}$$

Below we prove three assertions regarding the matrices just defined. These assertions make it possible to present our convergence analysis for the new algorithm compactly with alleviated notation.

**Lemma 1.** *Given $R \succeq 0$, let $\beta > 0$, $\mu, \rho \in (0, 1)$, and $\varrho \in (0, 1]$. The matrices $H, M$, and $Q$ defined, respectively, in Equations (15) and (16) have the relationships as follows:*

$$H \succ 0, \quad HM = Q \tag{17}$$

*and*

$$G := Q^\mathsf{T} + Q - M^\mathsf{T} H M \succeq 0. \tag{18}$$

**Proof.** We consider two cases.

(I). $\rho \in (0,1)$, $\varrho \in (0,1)$. We only need to check that

$$\bar{H} = \begin{pmatrix} R & 0 & 0 \\ 0 & \rho + \varrho - \rho\varrho & \rho \\ 0 & \rho & 1 \end{pmatrix} \succ 0. \tag{19}$$

Note that

$$\begin{cases} \rho + \varrho - \rho\varrho = \rho + \varrho(1 - \rho) > 0, \\ \rho + \varrho - \rho\varrho - \rho^2 = (\rho + \varrho)(1 - \rho) > 0. \end{cases}$$

Then we have

$$\bar{H} \succ 0. \tag{20}$$

For any $w = (x_1, z_2, \lambda) \neq 0$, since $\rho, \varrho \in (0,1)$, the assertion $H \succ 0$ is verified.
With the matrices $H$, $M$, and $Q$ at hand, we easily obtain

$$\begin{aligned} HM &= \begin{pmatrix} R & 0 & 0 \\ 0 & \frac{\rho+\varrho-\rho\varrho}{\rho+\varrho}\beta B^\mathsf{T}B & -\frac{\rho}{\rho+\varrho}B^\mathsf{T} \\ 0 & -\frac{\rho}{\rho+\varrho}B & \frac{1}{(\rho+\varrho)\beta}I_N \end{pmatrix} \begin{pmatrix} I_N & 0 & 0 \\ 0 & I_N & 0 \\ 0 & -\varrho\beta B & (\rho+\varrho)I_N \end{pmatrix} \\ &= \begin{pmatrix} R & 0 & 0 \\ 0 & \beta I_N & -\rho B^\mathsf{T} \\ 0 & -B & \frac{1}{\beta}I_N \end{pmatrix} = Q. \end{aligned} \tag{21}$$

The second assertion $HM = Q$ is proved. Consequently, we have

$$M^\mathsf{T}HM = M^\mathsf{T}Q = \begin{pmatrix} I_N & 0 & 0 \\ 0 & I_N & -\varrho\beta B^\mathsf{T} \\ 0 & 0 & (\rho+\varrho)I_N \end{pmatrix} \begin{pmatrix} R & 0 & 0 \\ 0 & \beta I_N & -\rho B^\mathsf{T} \\ 0 & -B & \frac{1}{\beta}I_N \end{pmatrix} \tag{22}$$

$$= \begin{pmatrix} R & 0 & 0 \\ 0 & (1+\varrho)\beta B^\mathsf{T}B & -(\rho+\varrho)B^\mathsf{T} \\ 0 & -(\rho+\varrho)B & \frac{\rho+\varrho}{\beta}I_N \end{pmatrix}. \tag{23}$$

Using Equations (15) and (16), and the above equation, we have

$$\begin{aligned} G &= Q^\mathsf{T} + Q - M^\mathsf{T}HM \\ &= \begin{pmatrix} 2R & 0 & 0 \\ 0 & 2\beta I_N & -(1+\rho)B^\mathsf{T} \\ 0 & -(1+\rho)B & \frac{2}{\beta}I_N \end{pmatrix} - \begin{pmatrix} R & 0 & 0 \\ 0 & (1+\varrho)\beta B^\mathsf{T}B & -(\rho+\varrho)B^\mathsf{T} \\ 0 & -(\rho+\varrho)B & \frac{\rho+\varrho}{\beta}I_N \end{pmatrix} \\ &= \begin{pmatrix} R & 0 & 0 \\ 0 & (1-\varrho)\beta B^\mathsf{T}B & (\varrho-1)B^\mathsf{T} \\ 0 & (\varrho-1)B & \frac{2-\rho-\varrho}{\beta}I_N \end{pmatrix}. \end{aligned}$$

Note that $\beta > 0$ and $\rho, \varrho \in (0,1)$. Thus, for any $w = (z_1, z_2, \lambda) \neq 0$, we have

$$\begin{cases} 1 - \varrho > 0 \\ (1-\varrho)(2-\rho-\varrho) - (1-\varrho)^2 = (1-\varrho)\{(1-\varrho) + (1-\rho)\} - (1-\varrho)^2 > 0 \end{cases}.$$

Therefore, the matrix $G$ is positive definite.

(II). $\rho \in (0, 1)$ and $\varrho = 1$. Note that

$$H = \begin{pmatrix} R & 0 & 0 \\ 0 & \frac{\beta}{\rho+1}B^\mathsf{T}B & -\frac{\rho}{\rho+1}B^\mathsf{T} \\ 0 & -\frac{\rho}{\rho+1}B & \frac{1}{(\rho+1)\beta}I_N \end{pmatrix}.$$

Thus, it is positive definite, and

$$G = \begin{pmatrix} R & 0 & 0 \\ 0 & 0 & 0 \\ 0 & 0 & \frac{1-\rho}{\beta}I_N \end{pmatrix}.$$

$G$ is only positive semi-definite. Here, we would emphasize that we do not require the positive definiteness of $G$. Instead, positive semi-definiteness of $G$ is enough for our algorithmic analysis. □

### 2.2. Algorithm

In this section, we will present our new algorithm to solve Equation (4). However, we first present the iterative scheme by using the standard strictly contractive Peaceman–Rechford splitting method with two different relaxation factors:

$$z_1^{k+1} = \arg\min_{z_1 \in \mathbb{R}^N} \mathcal{L}_\beta(z_1, z_2^k, \lambda^k), \tag{24a}$$

$$\lambda^{k+\frac{1}{2}} = \lambda^k - \rho\beta(Az_1^{k+1} - Bz_2^{k+1}), \tag{24b}$$

$$z_2^{k+1} = \arg\min_{z_2 \in \mathbb{R}^N} \mathcal{L}_\beta(z_1^k, z_2, \lambda^{k+\frac{1}{2}}), \tag{24c}$$

$$\lambda^{k+1} = \lambda^{k+\frac{1}{2}} - \varrho\beta(Az_1^{k+1} - Bz_2^{k+1}). \tag{24d}$$

By introducing a customized proximal term, especially for ultrasound imaging, our improved compressive deconvolution (ICD) method has the iterative scheme:

$$z_1^{k+1} = \arg\min_{z_1 \in \mathbb{R}^N} \mathcal{L}_\beta(z_1, z_2^k, \lambda^k) + \frac{1}{2}\| z_1 - z_1^k \|_R^2, \tag{25a}$$

$$\lambda^{k+\frac{1}{2}} = \lambda^k - \rho\beta(Az_1^{k+1} - Bz_2^{k+1}), \tag{25b}$$

$$z_2^{k+1} = \arg\min_{z_2 \in \mathbb{R}^N} \mathcal{L}_\beta(z_1^k, z_2, \lambda^{k+\frac{1}{2}}), \tag{25c}$$

$$\lambda^{k+1} = \lambda^{k+\frac{1}{2}} - \varrho\beta(Az_1^{k+1} - Bz_2^{k+1}), \tag{25d}$$

where $R = \begin{pmatrix} \frac{\beta}{\tau} - \beta\Psi^\mathsf{T}\Psi & 0 \\ 0 & \frac{\beta}{\gamma} - \beta H^\mathsf{T}H \end{pmatrix}$ is a customized semi-definite matrix. Note that the equivalence of Equations (25b)–(25d) and Equations (6c)–(6e) is evident, while the relationship between the closed-form solution expressed by Equation (25a) and Equations (6a) and (6b) is not evident. We illustrate the latter in Appendix A.

### 2.3. Global Convergence

To make the analysis more elegant, we reformulate ICD Equations (25a)–(25d) into the form

$$z_1^{k+1} = \arg \min_{z_1 \in \mathbb{R}^n} \{\theta_1(z_1) - (\lambda^k)^\mathsf{T} A z_1 + \frac{\beta}{2} \| A z_1 + B z_2^k \|^2 + \frac{1}{2} \| z_1 - z_1^k \|_R^2\} \tag{26a}$$

$$\lambda^{k+\frac{1}{2}} = \lambda^k - \rho\beta(A z_1^{k+1} + B z_2^k) \tag{26b}$$

$$z_2^{k+1} = \arg \min_{z_2 \in \mathbb{R}^n} \left\{\theta_2(z_2) - (\lambda^{k+\frac{1}{2}})^\mathsf{T} B z_2 + \frac{\beta}{2} \| A z_1^{k+1} + B z_2 \|^2\right\} \tag{26c}$$

$$\lambda^{k+1} = \lambda^{k+\frac{1}{2}} - \varrho\beta(A z_1^{k+1} + B z_2^{k+1}). \tag{26d}$$

Now we analyze the convergence for our proposed ICD method expressed by Equation (26). We prove its global convergence from the contraction perspective. In order to further alleviate the notation in our analysis, we define an auxiliary sequence $\tilde{w}^k$ as

$$\tilde{w}^k = \begin{pmatrix} \tilde{z}_1^k \\ \tilde{z}_2^k \\ \tilde{\lambda}^k \end{pmatrix} = \begin{pmatrix} z_1^{k+1} \\ z_2^{k+1} \\ \lambda^k - \beta(A z_1^{k+1} + B z_2^k) \end{pmatrix} \tag{27}$$

where $(z_1^{k+1}, z_2^{k+1})$ is produced by Equations (26a) and (26c), and we immediately have

$$z_1^{k+1} = \tilde{z}_1^k, \quad z_2^{k+1} = \tilde{z}_2^k, \qquad \lambda^{k+\frac{1}{2}} = \lambda^k - \rho(\lambda^k - \tilde{\lambda}^k),$$

and

$$\begin{aligned}
\lambda^{k+1} &= \lambda^{k+\frac{1}{2}} - \varrho\beta(A \tilde{z}_1^k + B \tilde{z}_2^k) \\
&= \lambda^k - \rho(\lambda^k - \tilde{\lambda}^k) - \varrho[\beta(A \tilde{z}_1^k + B z_2^k) - \beta B(z_2^k - \tilde{z}_2^k)] \\
&= \lambda^k - \rho(\lambda^k - \tilde{\lambda}^k) - \varrho[\lambda^k - \tilde{\lambda}^k - \beta B(z_2^k - \tilde{z}_2^k)] \\
&= \lambda^k - [(\rho + \varrho)(\lambda^k - \tilde{\lambda}^k) - \varrho\beta B(z_2^k - \tilde{z}_2^k)].
\end{aligned}$$

Moreover, we have the following relationship:

$$\begin{pmatrix} z_1^{k+1} \\ z_2^{k+1} \\ \lambda^{k+1} \end{pmatrix} = \begin{pmatrix} z_1^k \\ z_2^k \\ \lambda^k \end{pmatrix} - \begin{pmatrix} I_N & 0 & 0 \\ 0 & I_N & 0 \\ 0 & -\varrho\beta B & (\rho + \varrho) I_N \end{pmatrix} \begin{pmatrix} z_1^k - \tilde{z}_1^k \\ z_2^k - \tilde{z}_2^k \\ \lambda^k - \tilde{\lambda}^k \end{pmatrix},$$

which can be reformulated as a compact form under the notation of $w^k$ and $\tilde{w}^k$:

$$w^{k+1} = w^k - M(w^k - \tilde{w}^k), \tag{28}$$

where $M$ is defined in Equation (15).

Now we start to prove some properties for the sequence $\{\tilde{w}^k\}$ defined in Equation (27). We are interested in estimating how accurate the point $\tilde{w}^k$ is to a solution point $w^*$ of VI($F, \Omega, \theta$). The main result is proved in Theorem 1. Now, we try to find a lower bound in terms of the discrepancy between $\|w - w^{k+1}\|_H^2$ and $\|w - w^k\|_H^2$ for any $w \in \Omega$.

**Theorem 1.** *Let $\{w^k\}$ be generated by Equation* (26) *and let $\{\tilde{w}^k\}$ be defined in Equation* (27). *Let $H$ and $G$ be defined in Equations* (19) *and* (18), *respectively. Then, for any $w \in \Omega$, we have*

$$\theta(\tilde{u}^k) - \theta(u) + (\tilde{w}^k - w)^\mathsf{T} F(w) \le \frac{1}{2}(\|w - w^k\|_H^2 - \|w - w^{k+1}\|_H^2) - \frac{1}{2}\|w^k - \tilde{w}^k\|_G^2. \tag{29}$$

**Proof.** Refer to Appendix B. □

The next lemma demonstrates the contraction property of the sequence $\{w^k\}$ generated by Equation (26).

**Lemma 2.** *Let $\{w^k\}$ be generated by Equation (26) with $0 < \rho < 1$ and $0 < \varrho < 1$, and let $\{\tilde{w}^k\}$ be defined in Equation (27). Let $H$ and $G$ be defined in Equations (18) and (19), respectively. Then, for any $w^* \in \Omega^*$, we have*

$$\|w^{k+1} - w^*\|_H^2 \leq \|w^k - w^*\|_H^2 - \|w^k - \tilde{w}^k\|_G^2. \tag{30}$$

**Lemma 3.** *Let the sequence $\{w^k\}$ be generated by Equation (26) with $0 < \rho < 1$ and $\varrho = 1$. Then we have*

$$
\begin{aligned}
\|w^{k+1} - w^*\|_H^2 \quad \leq \quad & \|w^k - w^*\|_H^2 - \{\|z_1^k - \tilde{z}_1^k\|_R^2 \\
& + \frac{(3\rho+1)(1-\rho)}{1+\rho}\beta\|z_2^k - \tilde{z}_2^k\|^2 + (1-\rho)\beta\|\tilde{z}_1^k - \tilde{z}_2^k\|^2\}.
\end{aligned}
\tag{31}
$$

With the above lemmas, we can finally obtain the global convergence theorem of ICD method for solving $\text{VI}(\Omega, F, \theta)$ as follows:

**Theorem 2.** *The sequence $\{w^k\}$ generated by Equation (26) converges to some $w^\infty$ that is a solution of $\text{VI}(\Omega, F, \theta)$.*

## 3. Numerical Results

### 3.1. Numerical Simulations

#### 3.1.1. Simulated US Images

Two ultrasound data sets, named *group1* and *group2*, were obtained by 2D convolution between spatially invariant PSFs and the TRFs [15]. The results were quantitatively evaluated in terms of peak signal-to-noise ratio (PSNR), image structural similarity SSIM [18], improvement in SNR (ISNR), and the normalized root mean square error (NRMSE). The metrics are expressed as follows:

$$\text{PSNR} = 10\log_{10}\frac{NL^2}{\|x - \hat{x}\|^2}, \tag{32}$$

$$\text{SSIM} = \frac{(2\mu_x\mu_{\hat{x}} + c_1)(2\sigma_{x\hat{x}} + c_2)}{(\mu_x^2 + \mu_{\hat{x}}^2 + c_1)(\sigma_x^2 + \sigma_{\hat{x}}^2 + c_2)}, \tag{33}$$

$$\text{ISNR} = 10\log_{10}\frac{\|x - y\|^2}{\|x - \hat{x}\|}, \tag{34}$$

$$\text{NRMSE} = \sqrt{\frac{\|x - \hat{x}\|^2}{\|x\|^2}}, \tag{35}$$

where $x, y, \hat{x}$ are respectively the original image, the RF image, and the reconstructed image. $L$ denotes the maximum intensity value in $x$. $\sigma_x$ and $\sigma_{\hat{x}}$ are the mean and variance values of $x$ and $\hat{x}$; $c_1 = (k_1 C)^2$ and $c_2 = (k_2 C)^2$ are two variables stabilizing the division with a weak denominator. $C$ represents the dynamic range of the pixel-values and $k_1$, $k_2$ denote constants. Herein, $k_1 = 0.01$, $k_2 = 0.03$, and $C = 1$.

#### 3.1.2. In Vivo US Images

We consider two real in vivo US images [19]: (a) *Mouse bladder*: The observed image is with the size $400 \times 256$ as shown in Figure 1. The number of homogeneous regions was set to K = 3 in this experiment, which is sufficient to represent the anatomical structures of the image. (b) *Skin melanoma*: The second in vivo image (of size $400 \times 298$) represents a skin melanoma tumor, as shown in Figure 2.

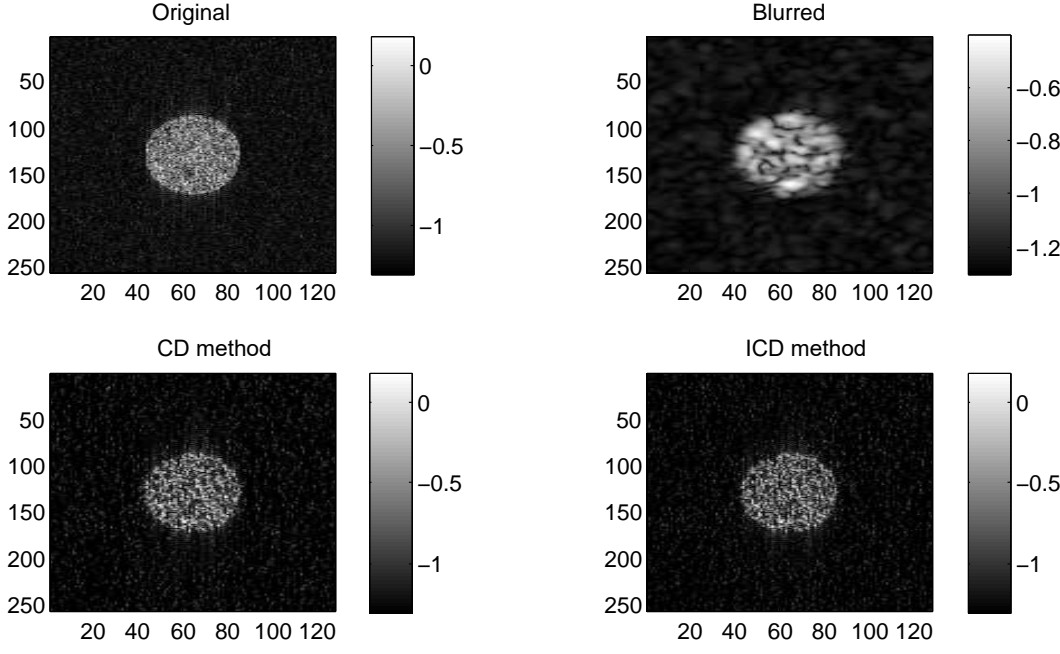

**Figure 1.** Results on *group1* with CS ratio = 0.6.

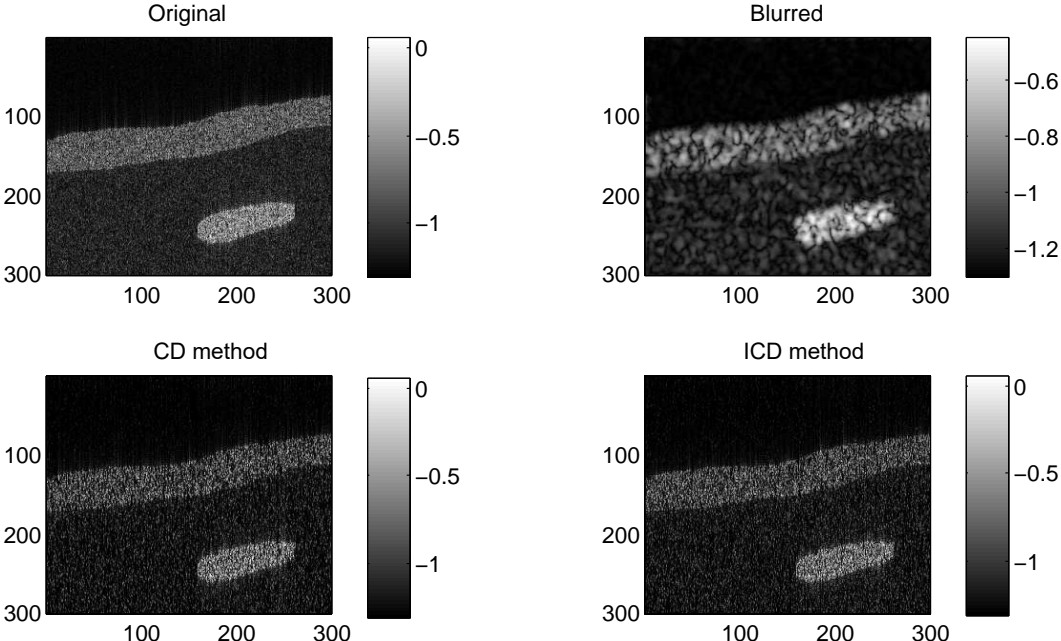

**Figure 2.** Results on *group2* with CS ratio = 0.6.

Since the ground truth of the TRF and the label map are not available for in vivo US data, the quality of the deconvolution results is evaluated using the contrast-to-noise ratio (CNR) [20]:

$$\text{CNR} = \frac{|\mu_1 - \mu_2|}{\sqrt{\sigma_1^2 + \sigma_2^2}}, \tag{36}$$

where $\mu_1$ and $\mu_2$ are the mean of pixels located in two regions extracted from the image, while and are the standard deviations of the same blocks. All code was written in MATLAB and performed on a ThinkPad computer equipped with Windows 7, 2.60 GHz and 2 GB of memory. Based on the same stopping criterion, $\frac{\|x^k - x^{k-1}\|}{\|x^{k-1}\|} < 10^{-3}$ is adopted.

*3.2. Results*

The quantitative results reported in Figures 1–8 confirm that, given the same maximum number of iteration (we set ItMax = $1 \times 10^4$), the ICD method can achieve better reconstruction quality gain than CD method for both simulated and real US images. Moreover, based on Figures 9 and 10, ICD method converges faster than the existing CD method for all CS ratios. We should remark that, for the *group2* data set, the metric of the SINR achieved via the CD method is negative. This may be because the CD method is unstable under low CS ratios.

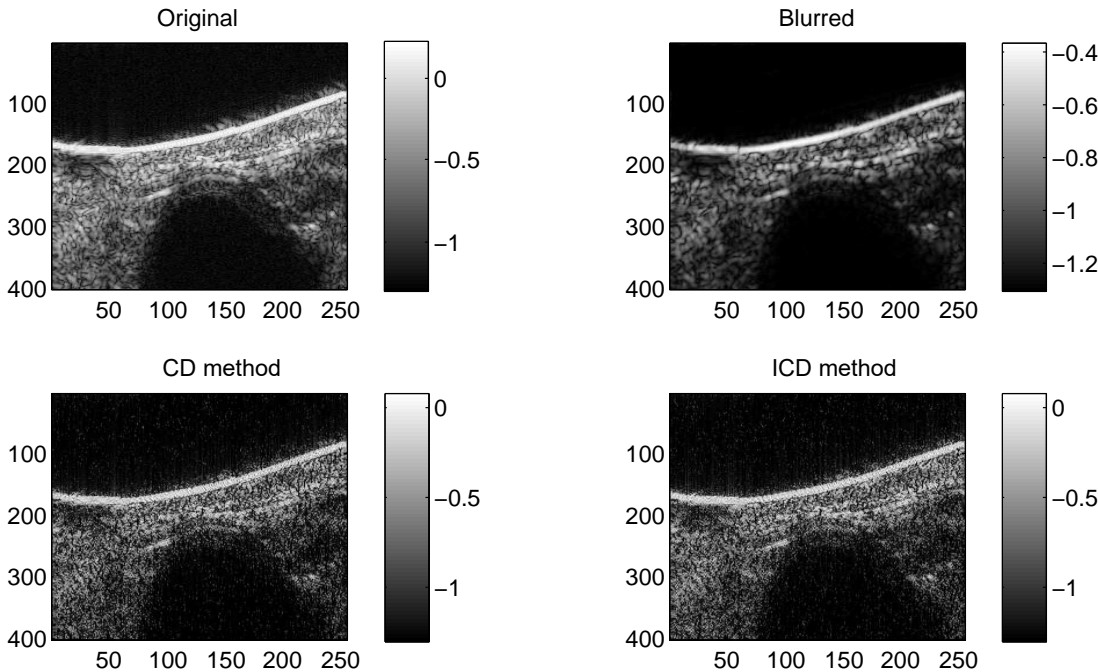

**Figure 3.** Results on *Mouse bladder* with CS ratio = 0.4.

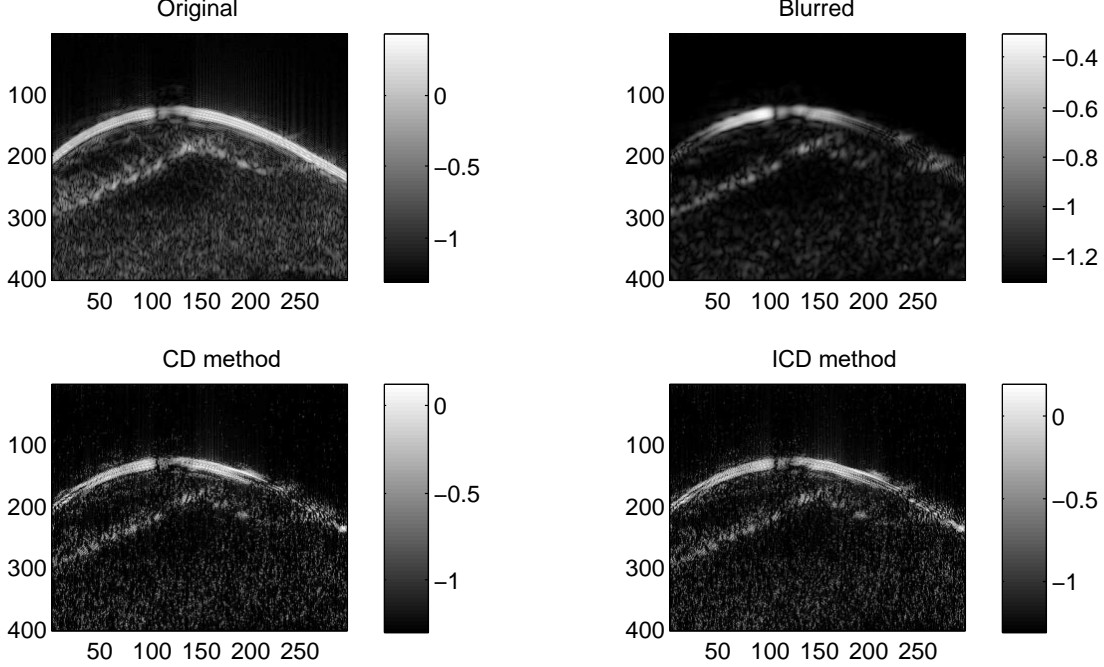

**Figure 4.** Results on *Skin melanoma* with CS ratio = 0.4.

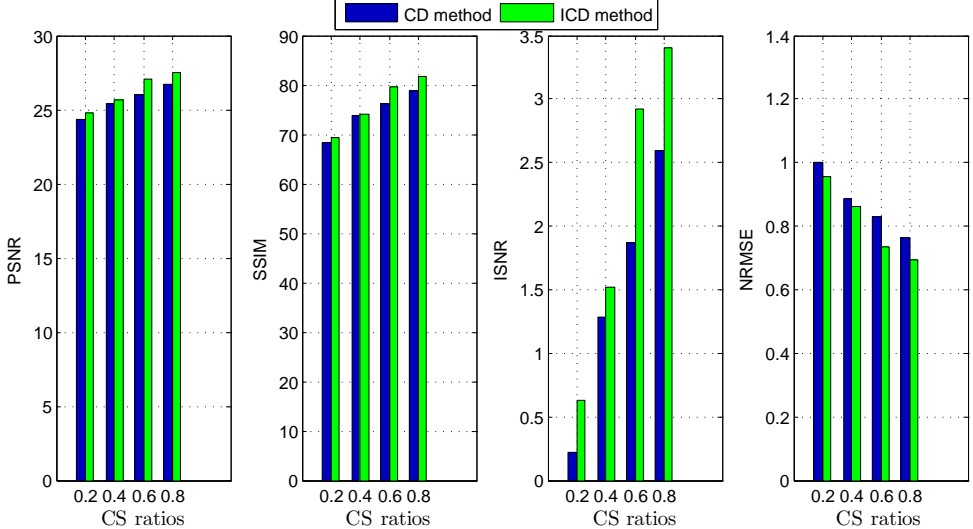

**Figure 5.** Deconvolution quality assessment for *group1*.

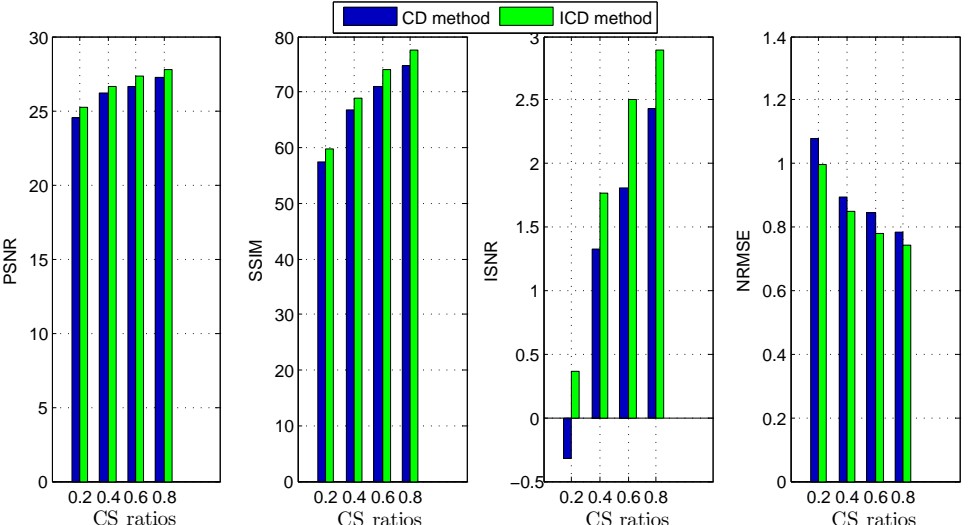

**Figure 6.** Deconvolution quality assessment for *group2*.

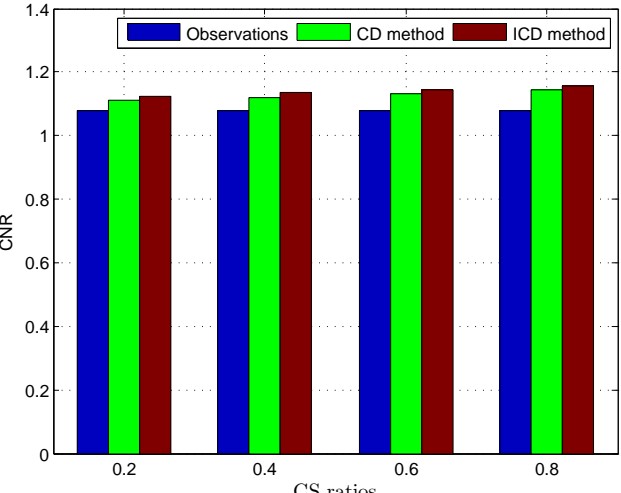

**Figure 7.** Deconvolution quality assessment for *Mouse bladder*.

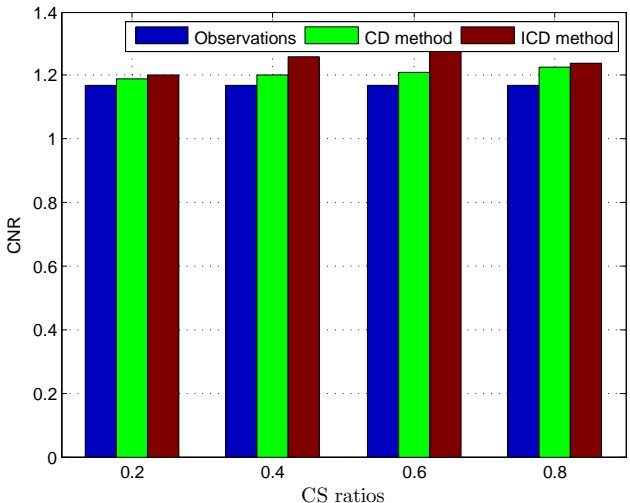

**Figure 8.** Deconvolution quality assessment for *Skin melanoma*.

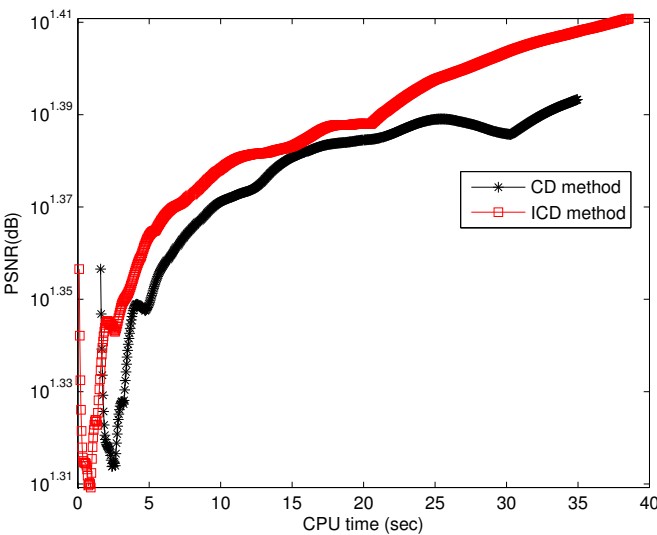

**Figure 9.** Running time for *group1*.

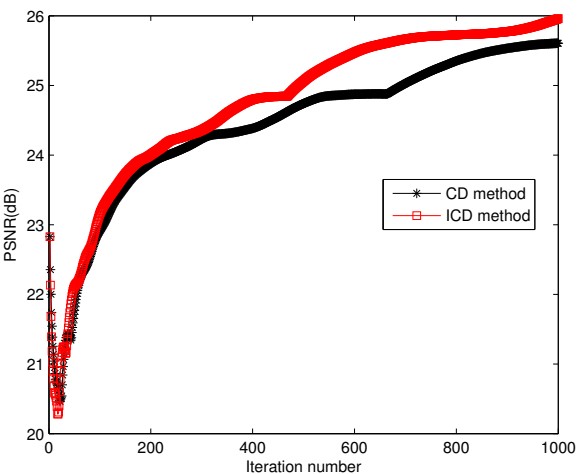

**Figure 10.** Iteration numbers for *group1*.

## 4. Discussions and Conclusions

In this paper, we proposed an improved compressive deconvolution method by introducing two different parameters in updating the dual variable to improve its convergence rate. We established the relationship between the two parameters under which we proved the global convergence of the algorithm. The ultrasound simulations show that the proposed method can achieve reconstruction US image with better quality gain under the same maximum number of iteration. Moreover, the ICD method has a much faster convergence rate compared with the conventional compressive deconvolution method. It should be noted that parameters such as $\rho$ and $\varrho$ are based only on empirical values, and a much greater deconvolution gain can be obtained if the parameters are adaptively optimized. This is the direction in which our research will continue.

**Author Contributions:** B.G. and S.X. proposed the architecture of ICD method. B.G. and X.L. derived the convergence analysis of ICD method. The simulations were implemented by B.G. and X.L. B.G. and X.L. wrote the manuscript. L.Z. and K.P. read and approved the final manuscript.

**Funding:** This work was supported in part by the National Natural Science Foundation of China under Grant 61673108, Grant 61571106, and Grant 71401176, and in part by the Natural Science Foundation of Jiangsu Province under Grant BK20170765.

**Conflicts of Interest:** The authors declare no conflict of interest.

## Appendix A. Proof of the Equivalence of Equations (25a) and (6a)–(6b)

**Proof.** From Equation (25a) and the definitions around Equation (7), it is not difficult to verify

$$
\begin{aligned}
w^{k+1} &= \arg\min_{w \in \mathbb{R}^n} \|w\|_1 - \lambda_2^k(v^k - \Psi w) + \frac{\beta}{2}\|\Psi w - v^k\|_2^2 + \frac{\beta}{2}\|w - w^k\|_{\frac{1}{\tau} - \Psi^\mathsf{T}\Psi}^2 \\
&= \arg\min_{w \in \mathbb{R}^n} \|w\|_1 + \frac{\beta}{2}\|\Psi w + \frac{\lambda_1^k}{\beta} - v^k\|_2^2 + \frac{\beta}{2}\|w - w^k\|_{\frac{1}{\tau} - \Psi^\mathsf{T}\Psi}^2 \\
&= \arg\min_{w \in \mathbb{R}^n} \|w\|_1 + \frac{\beta}{2\tau}\|w - w^k\|_2^2 - \frac{\beta}{2}\|\Psi w - \Psi w^k\|_2^2 + \frac{\beta}{2}\|\Psi w + \frac{\lambda_1^k}{\beta} - v^k\|_2^2 \\
&= \arg\min_{w \in \mathbb{R}^n} \|w\|_1 + \frac{\beta}{2\tau}\|w - w^k\|_2^2 + \beta\Psi^\mathsf{T}(\Psi w + \frac{\lambda_1^k}{\beta} - v^k)x \\
&= \arg\min_{w \in \mathbb{R}^n} \|w\|_1 + \frac{\beta}{2\tau}\|w - w^k + \tau\Psi^\mathsf{T}(\Psi w + \frac{\lambda_1^k}{\beta} - v^k)\|_2^2 \\
&= prox_{\tau\|\cdot\|_1/\beta}\left(v^k - \tau\Psi^\mathsf{T}(\Psi w + \frac{\lambda_1^k}{\beta} - v^k)\right).
\end{aligned}
$$

The above equation is exactly Equation (6a). Similarly,

$$
\begin{aligned}
x^{k+1} &= \arg\min_{x \in \mathbb{R}^n} \alpha\|x\|_p^p - \lambda_2^k(v^k - Hx) + \frac{\beta}{2}\|Hx - v^k\|_2^2 + \frac{\beta}{2}\|x - x^k\|_{\frac{1}{\gamma} - H^\mathsf{T}H}^2 \\
&= \arg\min_{x \in \mathbb{R}^n} \alpha\|x\|_p^p + \frac{\beta}{2}\|Hx + \frac{\lambda_2^k}{\beta} - v^k\|_2^2 + \frac{\beta}{2}\|x - x^k\|_{\frac{1}{\gamma} - H^\mathsf{T}H}^2 \\
&= \arg\min_{x \in \mathbb{R}^n} \alpha\|x\|_p^p + \frac{\beta}{2\gamma}\|x - x^k\|_2^2 - \frac{\beta}{2}\|Hx - Hx^k\|_2^2 + \frac{\beta}{2}\|Hx + \frac{\lambda_2^k}{\beta} - v^k\|_2^2 \\
&= \arg\min_{x \in \mathbb{R}^n} \alpha\|x\|_p^p + \frac{\beta}{2\gamma}\|x - x^k\|_2^2 + \beta H^\mathsf{T}(Hx + \frac{\lambda_2^k}{\beta} - v^k)x \\
&= \arg\min_{x \in \mathbb{R}^n} \alpha\|x\|_p^p + \frac{\beta}{2\gamma}\|x - x^k + \gamma H^\mathsf{T}(Hx + \frac{\lambda_2^k}{\beta} - v^k)\|_2^2 \\
&= prox_{\alpha\gamma\|\cdot\|_p^p/\beta}\left(x^k - \gamma H^\mathsf{T}(Hx^k + \frac{\lambda_2^k}{\beta} - v^k)\right).
\end{aligned}
$$

The above equation is exactly Equation (6b), so the proof is complete. $\quad\square$

## Appendix B. Proof of Theorem 1

To prove this main result, we require two lemmas. The first key lemma provides a lower bound on a specially constructed functional in terms of a quadratic term involving the matrix $Q$ defined in Equation (16) (It should be noted that the proof framework is based on He and Yuan's classical scheme [17]: First, carefully construct a semi-positive matrix (18) based on customized relaxation factors $\rho$ and $\varrho$ and then find the discrepancy between the current iterative point and the optimal point through their professional variational inequalities scheme (Inequality (A1)). The contractive property of iterative sequences is finally proven via their specially constructed identity. Since the proof line is straightforward and is similar to our previous work [16], we give convergence lemmas and theories without detailed proof.).

**Lemma A1.** *For a given $w^k \in \Omega$, let $w^{k+1}$ be generated by Equation* (26) *and let $\tilde{w}^k$ be defined in Equation* (27). *Then we have $\tilde{w} \in \Omega$ and*

$$\theta(u) - \theta(\tilde{u}^k) + (w - \tilde{w}^k)^\mathsf{T} F(\tilde{w}^k) \geq (w - \tilde{w})^\mathsf{T} Q(w^k - \tilde{w}^k), \quad \forall w \in \Omega \tag{A1}$$

*where $Q$ is defined in Equation* (16).

In the next lemma, we further analyze the right-hand side of Inequality (A1) and reformulate it as the sum of some quadratic terms. This new form is more convenient for our further analysis.

**Lemma A2.** *Let $w^k$ be generated by Equation* (26) *and let $\tilde{w}^k$ be defined in Equation* (27). *Let $Q$, $H$, and $G$ be defined in Equations* (19), (18), *and* (21), *respectively. Then, for any $w \in \Omega$, we have*

$$(w - \tilde{w}^k)^\mathsf{T} Q(w^k - \tilde{w}^k) = \frac{1}{2}(\|w - w^{k+1}\|_H^2 - \|w - w^k\|_H^2) + \frac{1}{2}\|w^k - \tilde{w}^k\|_G^2. \tag{A2}$$

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
