# Peer review of "Convergence Gain in Compressive Deconvolution: Application to Medical Ultrasound Imaging"

_applsci, doi:10.3390/app8122558_

Round 1
Reviewer 1 Report
Very interesting work and needed article in the field of ultrasound imaging. Please address the following issues.
1. Can the authors move some of the equations to Appendix
2. Please add scale bar and color bar to the figures 1 and 2
3. Use different markers for Figures 4 and 5
4. Add abit more discussion. What is the benefit of the proposed method over the current methods? Please compare it to :
Hojjatoleslami, S.A., Avanaki, M.R.N. and Podoleanu, A.G., 2013. Image quality improvement in optical coherence tomography using Lucy–Richardson deconvolution algorithm. Applied optics, 52(23), pp.5663-5670.
5. Add a few image quality metrics to compare the results.
Author Response
Responses to Reviewer’ Comments
Very interesting work and needed article in the field of ultrasound imaging. Please address the following issues.
Response:We would like to thank the reviewer for a positive comment on our paper.
1. Can the authors move some of the equations to Appendix
Response:Thanks for pointing this out. Some lemmas related to Theorem 2 are moved to Appendix B.
2. Please add scale bar and color bar to the figures 1 and 2
Response: Thanks for the comment. The scale bar and color bar were added.
3. Use different markers for Figures 4 and 5
Response: We acknowledge that the markers of Fig4 and Fig5 appears similar, mainly because the number of iterations is so many that markers are squeezed together.
4.Add abit more discussion. What is the benefit of the proposed method over the current methods? Please compare it to: Hojjatoleslami, S.A., Avanaki, M.R.N. and Podoleanu, A.G., 2013. Image quality improvement in optical coherence tomography using Lucy–Richardson deconvolution algorithm. Applied optics, 52(23), pp.5663-5670.
Response: Thanks for the comment. We would like to clarify that the implementation of Image quality improvement under customized deconvolution algorithm is outside the scope of the paper. The paper's focus is to accelerate the convergence rate of compressive deconvolution. We consider this suggestion to study how to apply Lucy–Richardson deconvolution algorithm for the ultrasound imaging problem in the future works. And we cite this method in the manuscript as an important deconvolution algorithm. As suggested, further discussions are added in Section 4.
5.Add a few image quality metrics to compare the results.
Response:Thanks for pointing this out. As suggested, we have added two quality metrics including the improvement in SNR (ISNR), and the normalized root mean square error (NRMSE) for simulated US image. Moreover,the CNR metric is introduced to evaluate the in vivo US data.

Reviewer 2 Report
The authors proposed a compressive deconvolution (CD) method based on the framework of strictly contractive Peaceman-Rechford splitting method. The paper is well-prepared and worth for ultrasound image processing.
In Figure 1, there are 5 images however, the explanation is only for 4 images. “From left to right: original ultrasound image, reconstruction results using YALL1, reconstruction results using ADMM and Reconstruction results using ICD method.” Figure 2 and 3 are same. Please refine those.
Author Response
Responses to Reviewer2’ Comments
Very interesting work and needed article in the field of ultrasound imaging. Please address the following issues.
The authors proposed a compressive deconvolution (CD) method based on the framework of strictly contractive Peaceman-Rechford splitting method. The paper is well-prepared and worth for ultrasound image processing.
Response:We would like to thank the reviewer for a positive comment on our paper.
1. In Figure 1, there are 5 images however, the explanation is only for 4 images. “From left to right: original ultrasound image, reconstruction results using YALL1, reconstruction results using ADMM and Reconstruction results using ICD method.” Figure 2 and 3 are same. Please refine those.
Response:Corrected. In fact,additional experiments containing important index like contrast-to-noise ratio(CNR) for evaluating images have been added.

Reviewer 3 Report
The paper deals with compressive deconvolution. To test their approach, authors used simulated images derived from medical ultrasound imaging.
The new approach is interesting but it constitutes again a new method among huge amount of methods with not impressive performances.
The mathematical details of the compressive deconvolution method are well presented. However, the structure of the proposed paper must be reviewed in regard to the main result. I suggest authors to make a new proposal inspired from my advices.
Indeed, if I understood correctly the improvement in the compressive deconvolution proposed by authors, the main result concerns the reduction of convergence time. The performances in terms of PSNR and SSIM being very similar for all techniques tested, these are not the main results. Notice that the main results should appear in a table; this is not the case in this paper since the secondary results appear in the table I. Note also that this reduction of convergence time should be useful for practical use but nothing is given in the paper.
From the previous paragraph the paper, through the introduction, the results and the discussion should be focalized on the main results that is the improvement of convergence time.
I suggest to authors to redo the structuration of their paper as:
1. Introduction
2. Method
2.1 Preliminaries a) Variational reformulation b) Notations
2.2 Algorithm : mathematical details sound well.
2.3 Convergence
3 Numerical Results: What is the real value of the noise in practice ? SNR=40dB is it realistic?
3.1 Numerical simulations: From my point of view there is a problem with the use of Field II simulator since the raw data seems to not be convolved with the PSF.
3.2 Results
4. Discussions and conclusion. Do not forget to discuss what can be the performances of the proposed approach with real images. What is the gain for medical images derived from ultrasound imaging? What are the drawbacks and advantages of the method?
Furthermore in regard to the content and results, the title of the article should be « Convergence gain in Compressive deconvolution : application to Medical ultrasound imaging ».
1. P2 line 19, “when p denotes ..” there is a conflict with the notation p line 19 and the vector p line20
2. P2 line 31 Meawhile = meanwhile
3. Points 1, 2, 3 p3 line 9 should be mentioned as objectives to solve your problem in the introduction
4. P3 line 23 Variatinal = variational
5. P8 line 26 According = according
6. P8 line 26, Please recall why it is a good think to divide the image in multiple areas. Do not forget that by doing this, you have to accept that the PSF should not be the same at different depth.
7. P9 Figures 1, 2 and p 10 Figure 3, the original image is not realistic, one does not see the impact of the PSF. The original image should be similar to the reconstructed image with YALL1. Where are the TRF image, the PSF, the PSF convolves with the TRF?
8. Page 10 Figure 4 and Figure 5, the scale (dB) should be the same for both graphs. Results derived from Fig4 and Fig 5 should be presented in a table since they are the main results
9. P11 Table 1 : why a two significative number?
10. P11 Conclusion : remove “experiment” and replace by simulations. In the conclusion a certain step back must appear, as well prospect must be focused on medical application, what is the gain for the clinician, the patient, … ?
11. P12 Ref 3 -> KOUAM ->KOUAME
Author Response
Responses to Reviewer2’ Comments
The paper deals with compressive deconvolution. To test their approach, authors used simulated images derived from medical ultrasound imaging.
The new approach is interesting but it constitutes again a new method among huge amount of methods with not impressive performances.
Response:We would like to thank the reviewer for a positive comment on our paper.
The mathematical details of the compressive deconvolution method are well presented. However, the structure of the proposed paper must be reviewed in regard to the main result. I suggest authors to make a new proposal inspired from my advices.
Indeed, if I understood correctly the improvement in the compressive deconvolution proposed by authors, the main result concerns the reduction of convergence time. The performances in terms of PSNR and SSIM being very similar for all techniques tested, these are not the main results. Notice that the main results should appear in a table; this is not the case in this paper since the secondary results appear in the table I. Note also that this reduction of convergence time should be useful for practical use but nothing is given in the paper.
Response:Thanks for the comment.We have added much more experiments for emphasizing the superiority of our faster ICD method. For example, we fixed the maximum number of iteration and show that ICD method can achieves a better reconstruction gain, which reflects ICD method’s faster convergence rate. And we also verify such conclusion in real in vivo US images.
From the previous paragraph the paper, through the introduction, the results and the discussion should be focalized on the main results that is the improvement of convergence time.
Response:Thanks for the comment.We greatly agree with your opinion.
I suggest to authors to redo the structuration of their paper as:
1. Introduction
2. Method
2.1 Preliminaries a) Variational reformulation b) Notations
2.2 Algorithm : mathematical details sound well.
2.3 Convergence
3 Numerical Results: What is the real value of the noise in practice ? SNR=40dB is it realistic?
Response:Thanks for the comment. SNR=40dB is only adopted in simulated US images like [R1]{R2}.
[R1] Z. Chen, A. Basarab, and D. Kouame,Compressive deconvolution in medical ultrasound imaging, IEEE Transactions on Medical Imaging,35 (2016), pp. 728–737.
{R2} N. Zhao, A. Basarab, D. Kouam, and J. Tourneret, Joint segmentation and deconvolution of ultrasound images using a hierarchical bayesian model based on generalized gaussian priors, IEEE Transactions on Image Processing, 25 (2016), pp. 3736–3750.
3.1 Numerical simulations: From my point of view there is a problem with the use of Field II simulator since the raw data seems to not be convolved with the PSF.
Response:Thanks for pointing this out. We reorganized the experimental parts. The real in vivo US images are considered, the evaluation of which is based on the contrast-to-noise ratio (CNR). For the simulated group1 and group2, it may be not convolved with the PSF via Field II simulator. Therefore, we just download the simulated US image including group1 and group2 from reference [R1].
[R1] Zhouye Chen, Adrian Basarab and Denis Kouam\'{e}, "Reconstruction of enhanced ultrasound images from compressed measurements using simultaneousdirection method of multipliers, IEEE Transactions on Ultrasonics, Ferroelectrics, and Frequency Control, 63(2016), pp. 1525–1534.
3.2 Results
4. Discussions and conclusion. Do not forget to discuss what can be the performances of the proposed approach with real images. What is the gain for medical images derived from ultrasound imaging? What are the drawbacks and advantages of the method?
Response:Thanks for pointing this out. two real in vivo US images including Mouse bladder and Skin melanoma are introduced. Like simulated US image, ICD-based reconstruction is much faster than that via CD method. Moreover, given fixed maximum number of iteration, ICD method can achieve better reconstruction gain which can be well depicted via CNR metric.
Furthermore in regard to the content and results, the title of the article should be ? Convergence gain in Compressive deconvolution : application to Medical ultrasound imaging ?.
Response:Thanks for your great idea. The title has been updated.
1. P2 line 19, “when p denotes ..” there is a conflict with the notation p line 19 and the vector p line20
Response:Corrected.
2. P2 line 31 Meawhile = meanwhile
Response:Corrected.
3. Points 1, 2, 3 p3 line 9 should be mentioned as objectives to solve your problem in the introduction
Response:Corrected.
4. P3 line 23 Variatinal = variational
Response:Corrected.
5. P8 line 26 According = according
Response:Corrected.
6. P8 line 26, Please recall why it is a good think to divide the image in multiple areas. Do not forget that by doing this, you have to accept that the PSF should not be the same at different depth.
Response:Thanks for point out this, such descriptions confusing readers have been omitted.
7. P9 Figures 1, 2 and p 10 Figure 3, the original image is not realistic, one does not see the impact of the PSF. The original image should be similar to the reconstructed image with YALL1. Where are the TRF image, the PSF, the PSF convolves with the TRF ?
Response:Thanks for point out this, some more visible Figures are updated. We turn to focus mainly on the reconstruction quality under the same iteration number, which also verify the faster convergence rate of proposed method.
8 and 9. Page 10 Figure 4 and Figure 5, the scale (dB) should be the same for both graphs. Results derived from Fig4 and Fig 5 should be presented in a table since they are the main results P11 Table 1 : why a two significative number?
Response:Thanks for point this out, we have replaced Table 1 by a few charts which can efficiently compare ICD method and CD method in a more intuitive way.
10. P11 Conclusion : remove “experiment” and replace by simulations. In the conclusion a certain step back must appear, as well prospect must be focused on medical application, what is the gain for the clinician, the patient, … ?
Response:Thanks for point this out, In conclusion part, we have updated as follows:
In this paper, we proposed an improved compressive deconvolution method by introducing two different parameters in updating the dual variable to improve its convergence rate. We established the relationship between the two parameters under which we proved the global convergence of the algorithm. The ultrasound simulations show that the proposed method can achieve reconstruction US image with better quality gain under the same maximum number of iteration. Moreover,ICD method has a much faster convergence rate compared with conventional compressive deconvolution method. Need to note, the parameters such as $\rho$ and $\varrho$ are based only on empirical values, and much more deconvolution gain could be obtained if the parameters are adaptively optimized. This is going to be our research direction.
11. P12 Ref 3 -> KOUAM ->KOUAME
Response:Thanks,it is corrected.

Round 2
Reviewer 3 Report
The paper is now in a publisable form.Thanks for your corrections